# Dietary Fat Intake and *KRAS* Mutations in Colorectal Cancer in a Moroccan Population

**DOI:** 10.3390/nu14020318

**Published:** 2022-01-13

**Authors:** Achraf El Asri, Karim Ouldim, Laila Bouguenouch, Mohammed Sekal, Fatima Zahra Moufid, Ellen Kampman, Inge Huybrechts, Marc J. Gunter, Sanae Abbaoui, Kaoutar Znati, Mehdi Karkouri, Khaoula El Kinany, Zineb Hatime, Meimouna Mint Sidi Deoula, Laila Chbani, Btissame Zarrouq, Karima El Rhazi

**Affiliations:** 1Laboratory of Epidemiology and Research in Health Sciences, Department of Epidemiology and Public Health, Faculty of Medicine and Pharmacy, Sidi Mohamed Ben Abdellah University, Fez 30000, Morocco; elkinanykhaoula@hotmail.fr (K.E.K.); dr.hatime.zineb@gmail.com (Z.H.); dolamouna@gmail.com (M.M.S.D.); btissame.zarrouq@usmba.ac.ma (B.Z.); karima.elrhazi@usmba.ac.ma (K.E.R.); 2Medical Genetics and Oncogenetics Unit, Hassan II University Hospital, Sidi Mohamed Ben Abdellah University, Fez 30000, Morocco; ouldim@yahoo.fr (K.O.); lbouguenouch@yahoo.com (L.B.); Fatimazohra.Moufid@usmba.ac.ma (F.Z.M.); 3Cancer Research Institute, Fez 20192, Morocco; 4Department of Anatomy and Cytopathology, Hassan II University Hospital, Sidi Mohammed Ben Abdallah University, Fez 30000, Morocco; mohammed.sekal@usmba.ac.ma (M.S.); laila.chbani@usmba.ac.ma (L.C.); 5Division of Human Nutrition and Health, Wageningen University and Research, 69000 Wageningen, The Netherlands; ellen.kampman@wur.nl; 6Nutrition and Metabolism Branch, International Agency for Research on Cancer, World Health Organization, 69372 Lyon, France; HuybrechtsI@iarc.fr (I.H.); GunterM@iarc.fr (M.J.G.); 7Faculty of Medicine and Pharmacy, Ibn Zohr University, Agadir 80035, Morocco; facoujda1@yahoo.fr; 8Department of Pathology, Ibn Sina University Hospital, Mohammed V University, Rabat 10001, Morocco; kaoutarznati@yahoo.fr; 9Pathologic Anatomy and Cytology Laboratory, Ibn Rochd University Hospital, Casablanca 20360, Morocco; mehdi.karkouri@gmail.com; 10Department of Biology and Geology, Teachers Training College (Ecole Normale Superieure), Sidi Mohamed Ben Abdellah University, Fez 30000, Morocco

**Keywords:** colorectal cancer, *KRAS* mutations, diet, fat intake, Moroccan population

## Abstract

Epidemiologic data support an association between diet and mutations in the Kirsten-ras (*KRAS*) gene involved in colorectal cancer (CRC) development. This study aimed to explore the associations between fat intake and *KRAS* mutations in codons 12 and 13 in cases of CRC in the Moroccan population. A multicenter case-series study nested in a large-scale Moroccan CRC case-control study was conducted. Among all CRC cases recruited, 151 specimens were available for the DNA mutation analysis. Logistic regression was used to calculate odds ratios (ORs) and 95% confidence intervals (Cis) for *KRAS* mutation status according to the fat intake variables. A *KRAS* mutation was detected in the CRC tumor of 34.4% of the patients among whom 65.4% had a single mutation at codon 12 and 34.6% had a single mutation at codon 13. Compared to low levels of consumption, a positive association was observed between high polyunsaturated fatty acids (PUFA) consumption (>16.9 g/day) and prevalence of *KRAS* mutations (OR = 2.15, 95% CI = 1.01–4.59). No statistically significant associations were observed for total fat, monounsaturated fatty acids, saturated fatty acids and *KRAS* mutations. The results of this study suggest that PUFA may be relevant in the etiology of CRC, possibly through the generation of G > A transitions at the *KRAS* oncogene. Further studies are needed to verify and explain this finding.

## 1. Introduction

Colorectal cancer (CRC), characterized by the accumulation of genetic and epigenetic alterations, is one of the most prevalent types of cancer worldwide [1]. Epidemiologic data support an association between CRC development and diet and lifestyle factors including body fatness, smoking, alcohol drinking, and diet [2,3].

Findings from the Word Cancer Research Fund/American Institute of Cancer Research (WCRF/AICR) report emphasizes the role of nutrition on CRC incidence. In fact, consumption of alcohol, red meat and processed meat increases CRC risk, while vegetables and fruits, dairy products, whole grains, dietary fiber, and calcium supplements are associated with lower CRC risk [4]. In Morocco, a Mediterranean country where adherence to the Mediterranean diet has become weaker in recent years [5], some studies have reported an increasing trend in CRC incidence. According to data from the cancer registry of greater Casablanca, the crude incidence of CRC increased markedly between 2005–2007 and 2012: it was estimated at 5.5 per 100,000 inhabitants in 2005–2007 [6] and increased to 8.1 per 100,000 inhabitants in 2008–2012 [7]. The growing incidence may be partially explained by the increased prevalence of obesity and by the ongoing nutritional transition [5,8]. The major change in dietary patterns includes a large increase in the consumption of foods high in sugar and fat, and thus high in calories, which may contribute to increasing body fatness. Indeed, daily fat supply per capita increased from 42 g to 59 g between 1968 and 1999 with a similar shift observed in caloric intake, increasing from 2466 to 2606 kcal per person [8]. According to the National Survey on Household Consumption and Expenditure, the consumption in quantity of fats increased between 2001 and 2014 following the intake of oils which increased on average by 5.4 L (22.4 L against 17.02 L per person per year) [9]. The daily food pattern in Morocco consisted of cereals and vegetables. Fresh fruits were consumed about 3 days/week followed by red meat (2 days/week), and fish and legumes (once a week) [5].

It has been long observed that dietary fat intake influences CRC incidence in animal models [10,11]. However, in humans, the results have been inconsistent [12,13]

One of the effects of high dietary fat intake is the exacerbation of obesity risk [14,15] which indirectly promotes CRC incidence [16,17]. In addition, certain prospective studies have shown that fat subtypes may be directly associated with CRC risk. Specifically, saturated fatty acids (SFA), monounsaturated fatty acids (MUFA) and cholesterol intake have been positively associated with CRC risk [18,19,20]. On the other hand, the association of polyunsaturated fatty acids (PUFA) and CRC seems to be complex. While the *n*-3 PUFAs have anti-neoplastic properties, the *n*-6 PUFAs may promote carcinogenesis [18,21,22,23,24,25,26].

In addition to the controversy regarding the association between fat intake andCRC risk, little is known about the relationship between fat consumption and genetic alterations involved in the carcinogenic process at the cellular level. In this context, most studies report that high-fat diets lead to higher colonic cell proliferation [11,27,28] and high intestinal excretion of bile acids, which are metabolized by gut bacteria into cancer-promoting agents [29]. In fact, a high fat diet has been shown to increase levels of deoxycholic acid (DCA) which can induce oxidative and nitrosative stress causing DNA damage and an increase in the frequency of mutations [30,31] in colonic epithelial cells.

Between 30 and 50% of colorectal tumors present with mutations in the Kirsten-ras (*KRAS*) gene and are recognized as an early molecular event in colorectal carcinogenesis [32]. A recent systematic review concluded there was conflicting evidence linking fat intake to the presence of *KRAS* mutations in colorectal tumors [33]. While some studies have found no association between fat intake and *KRAS* mutations [34,35,36,37], others have shown that high intake of polyunsaturated fat increases the risk of colon tumors harboring *KRAS* mutations [38,39]. In a study by Slattery et al. [40], individuals with high intake of dietary fat, saturated fat and monounsaturated fat were more likely to harbor a *KRAS* mutation, but the results were not statistically significant.

Currently, data regarding the link between dietary fat and *KRAS* mutations in colorectal cancer come primarily from developed countries [41,42], have many inconsistencies and there is a need to produce more evidence to establish a possible causal link between fat intake and the occurrence of *KRAS* mutations in CRC [33]. Thus, this study is aimed at exploring the associations between fat intake and *KRAS* mutation on codons 12 and 13 in the case of CRC risk in the Moroccan population.

## 2. Materials and Methods

### 2.1. Study Design

We conducted a multicenter case-series study nested in a large-scale Moroccan case-control study that took place between September 2009 and February 2017 and which aims to explore the behavioral, nutritional, and genetic risk factors of colorectal cancers in Morocco. Both genders, aged 18 years old and above and newly histological confirmed diagnosed patients were included in this study at the diagnosis or up to 3 months after diagnosis. Cases with hereditary cancer or with memory problems or seriously ill or confined to bed or those who are likely to have changed their diet at the onset of suspected symptoms before the CRC diagnosis were excluded. The full protocol of this study is detailed elsewhere [43]. Among all CRC cases recruited in the initial case control study, patients for whom specimens were available, were included in the genetic part of this study.

### 2.2. Ethical Procedure

This study was approved by the Ethics Committee at the University of Fez. Each CRC patient signed an informed consent form before participating in the study and gave permission to use his/her CRC embedded paraffin tissue sample for molecular analysis.

### 2.3. Data Collection

Data were collected in the four university hospitals through face-to-face interviews conducted by trained investigators. All participants who were newly diagnosed (no more than 3 months after the histological confirmation) were asked to answer questionnaires on the following topics: sociodemographic information; clinical data; substances use; physical activity levels and dietary data. The details of each topic were described previously [43].

Anthropometric measurements were also assessed and defined according to the global recommendations for health from the World Health Organization (WHO) [44]. Weight before diagnosis was self-reported and body mass index (BMI) was calculated as weight in kilograms divided by the square of height in meters (kg/m^2^). Tobacco use was defined in three classes (current smokers, ex-smokers, never smokers) according to the International Union against Tuberculosis and Lung Diseases [45], while alcohol consumption was classified into two categories (yes or no).

Physical activity intensity was assessed using the Global Physical Activity Questionnaire (GPAQ) and classified into three categories (high, moderate and low) based on the physical activity score expressed as metabolic equivalent of task-minutes per week (MET) [46]. The METs was divided into three categories: low intensity (<600 MET-minutes per week), moderate intensity (600–3000 MET-minutes per week), and vigorous intensity (≥3000 MET-minutes per week) [46]. Dietary data were obtained by a Moroccan, validated Food Frequency Questionnaire (FFQ) that includes 255 items comprising traditional Moroccan foods [47]. All cases were asked to report their habitual diet during the 12 months prior to the diagnosis in the validated FFQ [47].

The validation of the Moroccan FFQ used the FFQ European Global Asthma and Allergy Network is composed of 32 food sections and 200 food products. The foods representative of the Moroccan diet have been classified into 32 groups as follows: (1) bread, (2) breakfast with grains, (3) couscous, (4) pasta, (5) cake, (6) rice, (7) sugar, (8) sweets without chocolate, (9) chocolate, (10) vegetable oil, (11) margarine and vegetable fat, (12) butter and animals fat, (13) dried fruit, (14) legumes, (15) vegetables, (16) potatoes, (17) fruits, (18) juice, (19) non-alcoholic beverages, (20) coffee/tea, (21) beer, (22) wine, (23) other-alcoholic beverages, (24) red meat, (25) poultry, (26) sekat (offal), (27) fish, (28) eggs, (29) milk of cow/milk of soya, (30) cheese, (31) other dairy products, and (32) miscellaneous foods [47].

Frequency of dietary intake was estimated by choosing one of eight categories: never, once to three times per month, once a week, twice to four per week, five to six times per week, once per day, twice to three times, more than four times. Each food item was assigned a portion size using standard local household units to allow the calculation of intakes in grams/day. The nutrient and energy intake were calculated by multiplying the daily intakes of each food item by the nutrient and calorie content (per 100 g) of all food items using a Moroccan food composition table updated on 2020 [48]. Mean individual fat intakes (total FAT, PUFA, MUFA and SFA) per day were computed using the same computerized Moroccan food composition table.

Nutritional and lifestyle profile of all patients were recorded at the time of diagnosis confirmation.

For *KRAS* status data: The available tumor samples from CRC patients were collected between February 2016 and July 2017 from the archival collection of 4 major public health hospitals in Morocco. Tumor specimens from biopsies performed during diagnosis were embedded in paraffin and then analyzed by a pathologist; the molecular screening of the *KRAS* gene was performed at the genetics laboratory of CHU Hassan II of FEZ. DNA was isolated from unstained slides (containing at least 50% tumor cells) using Invitrogen RNA/DNA isolation kit. Exon 2 of the *KRAS* gene was PCR-amplified employing the following primers: F: 5′-GGTGGAGTATTTGATAGTGTA-3′ and R: 5′GGTCCTGCACCAGTAATATGCA-3′, then PCR products were screened for P12 and P13 codon mutations using direct Sanger sequencing. Thus, *KRAS* status was divided into two groups: *KRAS^+^* for the group with *KRAS* mutations and *KRAS*^−^ for the group without *KRAS* mutations.

### 2.4. Data Analysis

The Chi2-test was used to compare general sociodemographic characteristics and the student’s *t*-test was used to compare means of energy intake and the dietary intakes of total fat, PUFA, MUFA and SFA. A bivariate analysis was performed according to *KRAS* mutation status. Binary logistic regression was used to calculate ORs and 95% CIs for *KRAS* status according to the fat intake variables. We included in the initial model all the variables for which the *p*-value was less than 0.2 and we proceeded by step-by-step downward elimination until the final model was obtained [5]. The adjusted OR was calculated taking into account the potential confounders: age, sex and energy intake.

## 3. Results

Among 1483 cases recruited in the overall case-control study (48), mutation analysis was successful in 151 cases from 154 available specimens (98%). No statistical difference was shown between cases of the global case control study (*n* = 1483) and the cases included in this genetic part of the study (*n* = 151) neither for age (56.45 ± 13.98 and 53.6 ± 14 respectively), nor for gender (50.3% of women vs. 55% respectively), nor for energy intake (3244.66 ± 827.26 Kcal vs. 3288.87 ± 1004.44 Kcal respectively)

*KRAS* mutations were detected in the CRC tumors of 34.4% (52/151) of the patients, among whom 65.4% had a single mutation at codon 12 and 34.6% had a single mutation at codon 13. The most frequent mutations were: 35G > A (30.8%), 38G > A (30.8%) and 35G > T (15.4%) (Table 1). Among these mutations, 53.8% were located in the rectum, while 46.2% were found in the colon.

The general characteristics of the CRC patients with and without *KRAS* mutations in their tumors are shown in Table 2. Among the included CRC patients, 55% were female and the mean age was 53.66 ± 13.99 years. Three quarters were married (75.5%), 72.2% lived in urban areas, 62.3% were illiterate and 82.1% had a lower monthly income (less than 2000 MAD). Most participants were physically active (73.8%) and never smokers (84.8%). The prevalence of obesity (BMI ≥ 30 Kg/m^2^) in the sample was 16% and 43.3% were overweight. The tumor was located in the colon for 49.7% of patients and in the rectum for 50.33%. Compared to the group without *KRAS* mutations (*KRAS*^−^ group), the group carrying such mutations (*KRAS*^+^ group) showed a similar prevalence of different characteristics, with the exception of place of residence, where a high frequency of the *KRAS* mutation was recorded in rural areas, where 18 patients among 42 carried these mutation (prevalence = 43%) (Table 2).

The main food groups containing fat and assessed in the used FFQ were: vegetables oils (olive, argan, colza, sunflower, etc.), mixed oils used in cooking (extra virgin olive oil mixed with a fry oil), soya, animal butter, fat spread, traditional butter (smen), mayonnaise, walnuts and chick peas, almonds and double cream: SFA was mainly present in animal fat (52%), margarines and vegetable oils (35%), madeleine cake (16%), cream (15%), olive oil, chocolate (14%), corn oil, cheese (13%), croissants (12%), chips, red meat (11%). The main source of MUFA came from vegetable and mixed oils (56%), margarines (36%), animal fat (21%), mayonnaise (19%), dried fruit (16%), Moroccan sweets, cake, coconut (16%), chocolate (13%). PUFA is mainly found in: mayonnaise (43%), vegetable oils (30%), dried fruit (15%), chips, mixed fat (12%), soya beans (11%), potatoes cake (5%). No PUFA supplement intake was reported by the cases of this study.

The dietary intake in the *KRAS^+^* and *KRAS*^−^ groups are shown in Table 3. The *KRAS*^+^ group had higher intakes of energy, total fat, saturated fat, monounsaturated fat and polyunsaturated fat compared with the *KRAS*^−^ group. With the exception of saturated fat, all observed differences were statistically significant.

Based upon median intake of fat variables (low consumers versus high consumers) in the whole sample, no significant association was observed between total fat, SFA intakes and *KRAS* mutation.

PUFA and MUFA intakes were positively associated with tumors harboring *KRAS* mutations. For example, higher MUFA intake (beyond 34.9 g/day) was significantly associated with the *KRAS* mutations occurrence compared to the lowest intake (<34.9) (OR = 1.56, 95% CI = 1.17–3.61). An increase in the consumption of PUFA above 16.9 g/day was associated with an increase in the presence of *KRAS* mutations (OR = 2.48, 95% CI = 1.22–4.96) as compared to the reference group whose consumption was less than 16.9 g (Table 4).

After adjustment for confounding factors (age, sex, and energy intake), a positive association was observed between high PUFA consumption (>16.9 g/day) and *KRAS* mutations (OR = 2.15, 95% CI = 1.01–4.59) compared with the low consumption class (PUFA ≤ 16.9 g/day). However, no statistically significant associations were observed for total fat, MUFA, SFA and *KRAS* mutations (Table 4).

## 4. Discussion

In the present study, we aimed to determine the association between fat intake and the prevalence of *KRAS* mutations at codons 12 and 13 of exon 2 in colorectal carcinomas.

The *KRAS* mutation frequency (34.44%) at codons 12 and 13 of exon 2, found in this Moroccan sample, appears to be similar to patients from other regions of the world: Eastern Mediterranean Region (EMRO: 30.23%), European Region (EURO: 35.12%), Americas Region (PAHO: 31.83%), South-East Asian Region (SEARO: 33.17%), Western Pacific Region (WAPRO: 32.64%) [49] and Middle East and North Africa (MENA) region which includes Morocco (32.4%) [50]. When compared with national data, Dehbi et al., found a *KRAS* mutation prevalence equal to 39.5% (45/114) [51], when others observed respectively 29% (18/62), 23.9% (22/92), 33.3% (36/116) and 28% (16/57) which shows that this frequency is relatively similar [52,53,54].

Regarding *KRAS* mutation type, a particularly high distribution for both 38G > A (30.8%) and 35G > A (30.8%) was observed. This distribution differs from local and international data revealing a clear predominance of 35G > A [51,55,56,57]. More large-scale studies are needed to confirm our finding.

Among 151 patients, 65.4% who were positive for carriage of the *KRAS* mutation in their CRC tumor were from an urban area. However, *KRAS* mutation status was more frequent among cases from rural area (43%: 18/42) versus the urban area (31%: 34/109). This interesting result raises questions about the possibilities of changes in the genetic profiles of colorectal cancers in rural areas probably linked to changes in eating habits in these geographical areas as well.

Our study included the largest Moroccan series of tumor samples, which were collected from almost all regions in the country. These aspects should make our data more representative for the *KRAS* genetic profile of our population in comparison with other local studies [51,55,56,57].

We did not find any associations between the socio-demographic and clinical parameters, such as age, sex, BMI, tumor location, and *KRAS* mutations, a finding also observed in most national [54,55,56] and international studies [39,58,59].

Exploring energy and fat intake of CRC patients has shown that polyunsaturated fat (PUFA) but not, total fat, nor saturated and monounsaturated fats, was associated with increased risk of colon tumors with specific *KRAS* mutations. These findings are consistent with the results of studies conducted by Weijenberg et al., who reported the potential carcinogenic role of PUFAs, and especially linoleic acid [38,39]. The major difference in total energy intake between the *KRAS*^+^ and *KRAS*^−^ groups cannot be explained only by fat intake. In fact, the high energy intake is probably due to intake of carbohydrates, which contribute to 61% of the overall energy intake in Moroccan diet rich in carbohydrate [60].

Increasing evidence suggests that dietary PUFAs can interact with colonocytes either directly or indirectly by different mechanisms. The most important of which are: preferential uptake of PUFAs by the cancer cells; alteration of chromatin remodeling, membrane structure, downstream cell signaling and modifications of the epigenome via the modulation of DNA methylation [61,62]. Moreover, PUFAs can interact with oxygen radicals in biological systems inducing their peroxidation and the formation of a variety of products, many of which interact with proteins and DNA [63]. Malondialdehyde (MDA) is one of the most abundant products of PUFA peroxidation recognized by its carcinogenic and mutagenic effects [63]. The mutagenic capacity of this substance has been demonstrated in bacterial, mammalian and human cell assays and its carcinogenic power was established in a rodent model [64]. MDA induces insertions and deletions as well as base substitutions by reacting with DNA to form adducts to deoxyguanosine, deoxyadenosine, and deoxycytidine. The major adduct of DNA is a pyrimidopurinone, called M1G which is highly mutagenic in rodent and humans cells [63]. The most common mutations are G > T transversions and G > A transitions, with a minor contribution by G > C transversions [65,66]. In the current study, we observed that G > A (32/52: 62%) and G > T (8/52: 15%) are the most commonly represented mutations, which is consistent with the mutational spectrum caused by M1G. Although the role of PUFAs in DNA adduct formation are still not fully understood, researchers have observed that a high polyunsaturated fatty acid diet stimulated nearly a 20-fold increase in the levels of M1G up to levels of 28 per 10^7^ nucleotides [67].

This study has some potential limitations. First, because of the retrospective design of the study, nutritional collected data might be subject to recall bias, but since the study concerns only cases (no controls), the impact of this bias may be therefore attenuated. Secondly, patients may change their diet once the first suspected symptoms appeared before the cancer diagnosis is confirmed, which may lead to a misclassification bias. Including newly diagnosed patients (less than 3 months after diagnosis) and collecting habitual dietary intakes during the 12 months prior to the diagnosis may lessen this bias. In addition, participants were unaware of the potential risk of *KRAS* mutation. Thirdly, aberrations in other genes involved in CRC tumorigenesis are not available for this study, and specific PUFAs (ω-6 and ω-3 fatty acids) in relation to *KRAS* status, especially linolenic acid as the main source of ω-3 PUFAs, are not assessed. Fourthly, for this case series study, results are based on relatively small numbers of patients, which did not allow a stratified tumor location analysis, and only the most representative codons (12 and 13) were analyzed while the other codons (59, 61, 117, 146) could not be studied. Finally, the study was focused solely on fat intake to explain the variance between *KRAS*^+^ and *KRAS*^−^ groups, while other factors (not studied in the current study and worth further studies) such as carbohydrate could provide additional clarification on this variance, especially the difference in the energy intake between the two groups.

On the other hand, the study has several strengths. This is the first study in Africa attempting to elucidate the link between fat intake, considering different fat types, and *KRAS* mutations in colorectal cancer. In addition, participants were recruited from the main four large university hospitals centers (UHCs) in Morocco and could, therefore, be considered as representative of the Moroccan CRC population. The results obtained are reliable, since nutritional data collection was based on a FFQ adapted to the national context taking into account the local cultural specificities of all the included regions [47], fat variables extracted from a Moroccan food composition table updated in 2020. While the last version dating from 1984 does not take into account the dietary changes during the last two decades and does not include most of the food items included in the updated version [48], we estimate that the closed food composition table to what study participants have actually consumed is the updated version. Finally, all the genetic analyses of tissue samples were carried out in a single laboratory, to standardize the molecular analysis conditions as much as possible.

## 5. Conclusions

The results of this study suggest that poly-unsaturated fatty acids (PUFA) may be relevant in the etiology of CRC by potentially generating G > A transitions in the *KRAS* oncogene. Further studies are needed to verify and explain this finding. Possibly, PUFAs may also interact with other molecular pathways in colorectal cancer. Elucidating gene-diet interactions may help us better understand the mechanisms by which dietary factors affect risk of CRC, and consequently improving our understanding of CRC etiology.

## Figures and Tables

**Table 1 nutrients-14-00318-t001:** Distribution of *KRAS* mutations in codons 12 and 13 of exon 2 (N = 52).

Codon	Base	Wild Type	Mutant	AA Change	*KRAS* Mutations *n* (%)
12	1st	G	A	G12S	1 (1.9)
			T	G12C	3 (5.8)
	2nd	G	A	G12D	16 (30.8)
			T	G12V	8 (15.4)
			C	G12A	4 (7.7)
	3rd	T	A	No change	1 (1.9)
			C	No change	1 (1.9)
13	2nd 3rd	G	A	G13D	16 (30.8)
		C	T	No change	2 (3.8)

G: glycine; S: serine; C: cycteine; D: aspartic acid; V: valine; A: alanine.

**Table 2 nutrients-14-00318-t002:** General characteristics of the overall study population and the colorectal cancer cases stratified by the presence of *KRAS* mutations (N = 151).

General Characteristics		Whole Population *n* (%)	Colorectal Cancer		*p*
			With *KRAS* Mutation N = 52 *n* (%)	Without *KRAS* Mutation N = 99 *n* (%)	
**Age (Years), Mean (SD)**		53.6 (14)	54.6 (13.2)	53.2 (14.4)	0.55
Sex	female	83 (55.0)	28 (53.8)	55 (55.6)	0.84
Marital status	single	13 (8.6)	4 (7.7)	9 (9.1)	0.63
	married	114 (75.5)	37 (71.2)	77 (77.8)	
	divorced	7 (4.6)	3 (5.8)	4 (4.0)	
	widow	17 (11.3)	8 (15.4)	9 (9.1)	
Area of residency	urban	109 (72.2)	34 (65.4)	75 (75.8)	0.17
	rural	42 (27.8)	18 (34.6)	24 (24.2)	
Educational level	illiterate	94 (62.3)	33 (63.5)	61 (61.6)	0.48
	primary	28 (18.5)	9 (17.3)	19 (19.2)	
	secondary or higher	29 (19.2)	10 (19.2)	19 (19.2	
Monthly income (MAD)	≤2000	124 (82.1)	45 (86.5)	79 (79.8)	0.29
	>2000	27 (17.9)	7 (13.5)	20 (20.2)	
Smoking status	never smokers	128 (84.8)	44 (84.6)	84 (84.8)	0.82
	smokers and ex-smokers	23 (15.2)	8 (15.4)	15 (15.2)	
BMI	normal	61 (40.7)	20 (38.5)	41 (41.8)	0.90
	overweight	65 (43.3)	23 (44.2)	42 (42.9)	
	obesity	24 (16.0)	9 (17.3)	15 (15.3)	
Physical activity	inactive	39 (26.2)	12 (23.1)	27 (27.8)	0.52
	active	110 (73.8)	40 (76.9)	70 (72.2)	
Tumor location	colon	75 (49.7)	24 (46.2)	51 (51.5)	0.53
	rectum	76 (50.3)	28 (53.8)	48 (48.5)	
Alcohol consumption	yes	4 (2.6)	1 (1.9)	3 (3)	NS
	no	147 (97.7)	51 (98.1)	96 (97)	

BMI: body mass index; MAD: Moroccan dirham; SD: standard deviation; NS: non-significant.

**Table 3 nutrients-14-00318-t003:** Dietary fat intake stratified by the presence of *KRAS* mutations in codons 12 and 13.

	Group with *KRAS* Mutation Mean ± SD	Group without *KRAS* Mutation Mean ± SD	*p*
Energy intake (kcal/day)	4663.84 ± 1566.51	2853.73 ± 1402.67	0.03
Total fat (g/day)	116.11 ± 60.94	95.71 ± 35.49	0.01
Saturated fat (g/day)	45.71 ± 41.07	37.15 ± 34.80	0.18
MUFA (g/day)	43.82 ± 22.22	35.45 ± 18.09	0.01
PUFA (g/day)	19.77 ± 8.25	16.16 ± 5.76	0.01

MUFA: monounsaturated fatty acids; PUFA: polyunsaturated fatty acids; SD: standard deviation.

**Table 4 nutrients-14-00318-t004:** Crude and adjusted ORs for the risk of *KRAS* mutations in colorectal tumors, according to the intake of fat variables (low consumers versus high consumers based upon median intake) (N = 151).

Fat Intake		Lower Intake	Higher Intake
Total fat (g/day)	Median	≤94.04	>94.04
	OR (95% CI)	1	1.34 (0.68–2.64)
	aOR (95% CI)	1	1.35 (0.29–1.84)
Saturated fat (g/day)	Median	≤27.35	>27.35
	OR (95% CI)	1	1.61 (0.80–3.21)
	aOR (95% CI)	1	1.33 (0.29–2.42)
MUFA (g/day)	Median	≤34.92	>34.92
	OR (95% CI)	1	1.56 (1.17–3.61)
	aOR (95% CI)	1	1.11(0.54–2.24)
PUFA (g/day)	Median	≤16.87	>16.87
	OR (95% CI)	1	2.48 (1.22–4.96)
	aOR (95% CI)	1	2.15(1.01–4.59)

aOR: binary logistic regression adjusted OR for age, sex, energy intake.

## Data Availability

The datasets used and analyzed during the current study are available from the corresponding author on reasonable request.

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
