# Peer review of "Dietary Fat Intake and KRAS Mutations in Colorectal Cancer in a Moroccan Population"

_nutrients, 2022, doi:10.3390/nu14020318_

Round 1

Reviewer 1 Report

The study by Asri and colleagues demonstrates an association of the prevalence of KRAS mutations with poly-unsaturated fatty acid intake in a Moroccan population. Around 150 patients of which material was available for sequencing of KRAS had been selected from a larger case control study. After adjusting for various confounders, PUFA intake was identified as risk factor for KRAS mutations. The study is generally well designed and the conclusions are supported by the results. As a control population is missing, no direct conclusions can be drawn on whether PUFAs are generally associated with a higher risk for developing CRC, which limits the significance of the study to some extent. However, this is an innate feature of this study design and correctly acknowledged by the authors.

The study would benefit from English language editing but it otherwise clear and well written. I only have a few additional comments, which should be addressed by the authors prior to publication.

  • Lines 65 etc.: Are there no more recent data available than 2012 for CRC incidence or 1999 for fat supply? The latter number is already 23 years old and I image that things will have changed even further in the meantime.
  • Line 139: please define the abbreviation MET at its first occurrence.
  • Table 2: I think there may be a mix up of the columns or column names here. For example, it says 55 (55.6%) of the KRAS- patients were female, but in the heading it says total N=52. R another example, it says 77 were married but there are only 52 in the group. I thus would guess that the headings are swapped by mistake and that KRAS+ in table 2 is actually KRAS- and vice versa (or only the “N=” was swapped”).
  • Table 2: the naming of the groups is actually quite confusing here, possibly also for the reason of the mix-up explained above. Could you please add a specific sentence in the text and table legend which group you call KRAS+ and KRAS-? This is especially important as the interpretations of the other tables etc. also depend on this being unequivocally clear.
  • Lines 277-278: since you do not city any other, smaller, studies, I would delete this sentence

Author Response

The study by Asri and colleagues demonstrates an association of the prevalence of KRAS mutations with poly-unsaturated fatty acid intake in a Moroccan population. Around 150 patients of which material was available for sequencing of KRAS had been selected from a larger case control study. After adjusting for various confounders, PUFA intake was identified as risk factor for KRAS mutations. The study is generally well designed and the conclusions are supported by the results. As a control population is missing, no direct conclusions can be drawn on whether PUFAs are generally associated with a higher risk for developing CRC, which limits the significance of the study to some extent. However, this is an innate feature of this study design and correctly acknowledged by the authors.

The study would benefit from English language editing but it otherwise clear and well written. I only have a few additional comments, which should be addressed by the authors prior to publication.

Point 1: Lines 65 etc.: Are there no more recent data available than 2012 for CRC incidence or 1999 for fat supply? The latter number is already 23 years old and I image that things will have changed even further in the meantime.

Response 1: Thank you very much for pointing this out, but unfortunately no updates to the Moroccan registries have been published since 2012. Concerning studies that focus on the Moroccan food profile, in most cases they do not target the general population. So, among the rare studies that have measured dietary fat intake, we can mention the reference below which was included in the manuscript. Please see lines from 72 to 75, and reference number 10.

Maaroufi Y. National survey of household consumption and expenditure. Institutional site of the High Commission for Planning of the Kingdom of Morocco. [cited 29 déc 2021]. Available on: https://www.hcp.ma/Enquete-nationale-sur-la-consommation-et-les-depenses-des-menages_a95.html

Point 2: Line 139: please define the abbreviation MET at its first occurrence.

Response 2: As suggested by the reviewer, the definition of the abbreviation MET has been added at its first occurrence and also to the abbreviation section. Please see lines 145 and 372.

Point 3: Table 2: I think there may be a mix up of the columns or column names here. For example, it says 55 (55.6%) of the KRAS- patients were female, but in the heading it says total N=52. R another example, it says 77 were married but there are only 52 in the group. I thus would guess that the headings are swapped by mistake and that KRAS+ in table 2 is actually KRAS- and vice versa (or only the “N=” was swapped”).

Response 3: We apologize for any confusion and thank the reviewer for noticing this mistake in Table 2. As the reviewer mentioned, the “N=” was swapped. The mistake has been corrected. Please see Table 2.

Point 4: Table 2: the naming of the groups is actually quite confusing here, possibly also for the reason of the mix-up explained above. Could you please add a specific sentence in the text and table legend which group you call KRAS+ and KRAS-? This is especially important as the interpretations of the other tables etc. also depend on this being unequivocally clear.

Response 4: We thank the reviewer for asking about clarification related to clear up the ambiguity of the terms KRAS + and KRAS-. We have clarified the meaning of these terms in text and tables. Please see lines190,191,221,222 and tables 2 and 3.

Point 5: Lines 277-278: since you do not city any other, smaller, studies, I would delete this sentence

Response 5: Thank you very much for pointing this out. The local studies which included a small sample were recalled in the manuscript. Please see line 293, and references number 55, 56, 57, 58, and 59.

Reviewer 2 Report

This manuscript is the content of research on the relationship between dietary fat intake and KRAS mutations in colorectal cancer in Morocco. Interesting, but I have some comments to the author.

Comment 1. The expressions KRAS- and KRAS + are used in the text and Table 2. Does KRAS + mean Kras mutation negative and KRAS- indicates mutation positive? It is very difficult to understand because there is no explanation for grouping. Since KRAS mutation is described as N = 52 in Table 1, I think that KRAS + in Table 2 indicates Kras mutation negative and KRAS- indicates mutation positive. Is this understanding correct?

Comment 2. Although one line is described in "Limitation", KRAS mutations include codon 59, 61, 117, 146 as major mutations in addition to codon 12, 13. Is it okay to define KRAS mutations by validating only codon 12, 13? 

Comment 3. In Table 3, there is an average difference of about 1600 kcal between KRAS + vs KRAS - group in Energy intake, but there is only about 10 g / day (= 90 kcal / day) in Total fat. Furthermore, regarding PUFA, there is no difference of about 4 g / day between the two groups, and the KRAS + group consumes more. As mentioned in Comment 1, if KRAS + means Kras mutation negative group, it may be that people without KRAS mutation are taking more Fat and PUFA.

Also, is the difference in PUFA intake of about 4g / day really meaningful?

Rather, the difference in energy intake is remarkable, and the difference in the intake of calorie sources other than fat may be the problem.

Author Response

This manuscript is the content of research on the relationship between dietary fat intake and KRAS mutations in colorectal cancer in Morocco. Interesting, but I have some comments to the author.

Point 1. The expressions KRAS- and KRAS + are used in the text and Table 2. Does KRAS + mean Kras mutation negative and KRAS- indicates mutation positive? It is very difficult to understand because there is no explanation for grouping. Since KRAS mutation is described as N = 52 in Table 1, I think that KRAS + in Table 2 indicates Kras mutation negative and KRAS- indicates mutation positive. Is this understanding correct?

Response 1: We apologize for any confusion and thank the reviewer for noticing this error in Table 2. In fact, the “N=” was swapped in Table 2. We have rectified this error in Table 2, and we have clarified the meaning of “KRAS+ and KRAS- in the text. We have also replaced the ambiguous word “KRAS+” with “Group with KRAS mutation”, and “KRAS-“ with “Group without KRAS mutation” in Tables. Please see lines 190,191, 221,222 and Tables 2 and 3.

Point 2. Although one line is described in "Limitation", KRAS mutations include codon 59, 61, 117, 146 as major mutations in addition to codon 12, 13. Is it okay to define KRAS mutations by validating only codon 12, 13? 

Response 2: We would like to thank the reviewer for raising this important point. In fact, codons 12 and 13 are the most affected by KRAS mutations compared to the others codons which are negligible (Please see the reference below). That’s why, most of the articles focus mainly on these two codons to describe the KRAS status. Consequently, we think that codons 12 and 13 are well representative of the most frequent mutations in this gene.

As suggested by the reviewer, this important point was added to the “limitations of the study”. Please see lines from 334 to 336.

Reference : Cox AD, Der CJ. Ras history. Small GTPases. 2010;1(1):2‑27.

https://www.ncbi.nlm.nih.gov/pmc/articles/PMC3109476/

Point 3: In Table 3, there is an average difference of about 1600 kcal between KRAS + vs KRAS - group in Energy intake, but there is only about 10 g / day (= 90 kcal / day) in Total fat. Furthermore, regarding PUFA, there is no difference of about 4 g / day between the two groups, and the KRAS + group consumes more. As mentioned in Comment 1, if KRAS + means Kras mutation negative group, it may be that people without KRAS mutation are taking more Fat and PUFA.

Also, is the difference in PUFA intake of about 4g / day really meaningful?

Rather, the difference in energy intake is remarkable, and the difference in the intake of calorie sources other than fat may be the problem.

Response 3: Thank you so much for your valuable comments. Following the reviewer’s comment, we would like to mention that the data from a large perspective cohort study conducted by Matty P. Weijenberg (Please see the reference below), suggest that PUFA intake (linoleic acid intake) was strongly associated with colon tumors with an aberrant KRAS gene. In this study, the difference between the groups representing the quartiles was just around 4 grams (PUFA (g/day): Q1-11.6; Q2-16.0; Q3-20.9; Q4-29.3), and the PUFA intake median in Weinberg study was 16 g/day, which is very close to the result obtained in our study (16,87 g/day). Thereby, we think that the differences in PUFA intake between groups is meaningful.

Reference: Weijenberg MP, Lüchtenborg M, de Goeij AFPM, Brink M, van Muijen GNP, de Bruïne AP, et al. Dietary fat and risk of colon and rectal cancer with aberrant MLH1 expression, APC or KRAS genes. Cancer Causes Control. oct 2007;18(8):865‑79.

https://pubmed.ncbi.nlm.nih.gov/17636402/

On the other hand, even if there is a big difference in total energy intake, this factor is not established (until now) as being a risk neither for the development of the colorectal cancer nor for the mutations.

The high energy intake is mainly due to carbohydrates which contribute with 61% of the overall energy intake in Moroccan diet rich in carbohydrate (Please see the reference below). However, carbohydrates do not constitute a risk factor for CRC or mutations either. In addition, the OR adjusted for energy showed that the difference in PUFA is still significant.

A paragraph has been added to evoke this point. Please see lines from 302 to 306 and reference number 52.

Reference: Mziwira M, Ayachi ME, Lairon D, Belahsen R. Dietary habits of a Mediterranean population of women in an agricultural region of Morocco. Afr J Food Agric Nutr Dev. 11 juin 2015;15(2):9807‑10.

https://www.ajol.info/index.php/ajfand/article/view/118280

Reviewer 3 Report

Thank you for the opportunity to review this manuscript on dietary fat intake and KRAS mutation in colorectal cancer patients in Morocco. The study population is unique and this study is a nice addition to the existing scientific evidence on this topic. The manuscript is well-written and I have a few comments to clarify and strengthen this manuscript as follows:

Lines 223-231: The uniqueness of this study is its study population and location. That being said, readers may not be very familiar with Moroccan diet, although it says to be a Mediterranean diet. It is helpful to include more description/information on Moroccan diets and, for example, a table with major food items consumed within each food group (e.g., top 3 most consumed food items).

Line 223: What are blended and average oils?

Lines 227-229: For the “vegetable and blended oils”, do you mean vegetable oils and blended oils? I see vegetable fats listed separately. Please clarify. Also, if you can provide examples of vegetable oils , blended oils, and vegetable fats, that would be helpful.

Lines 262-263: Not all readers are familiar with these abbreviated region names. Please spell out or add more common names.

Lines 309-321: Another potential source of nutrients is supplements. Please comment on the prevalence of supplement use in this region and on the decision to not collect this data.

Lines 326-329: FFQ was administered in 2009 to 2017 (lines 109-110). Using updated version of 2020 food composition table (lines 326-329) may not reflect what study participants have actually consumed due to changes in food supply. Please justify the use of 2020 in your study.

Author Response

Thank you for the opportunity to review this manuscript on dietary fat intake and KRAS mutation in colorectal cancer patients in Morocco. The study population is unique and this study is a nice addition to the existing scientific evidence on this topic. The manuscript is well-written and I have a few comments to clarify and strengthen this manuscript as follows:

Point 1: Lines 223-231: The uniqueness of this study is its study population and location. That being said, readers may not be very familiar with Moroccan diet, although it says to be a Mediterranean diet. It is helpful to include more description/information on Moroccan diets and, for example, a table with major food items consumed within each food group (e.g., top 3 most consumed food items).

Response 1: Thank you so much for raising this concern. The presentation of the most consumed foods will shed light on the diet of our population, that’s why we have added a paragraph that describes the most consumed foods in Morocco. Please see lines from 75 to 78.  

Point 2: Line 223: What are blended and average oils?

Lines 227-229: For the “vegetable and blended oils”, do you mean vegetable oils and blended oils? I see vegetable fats listed separately. Please clarify. Also, if you can provide examples of vegetable oils, blended oils, and vegetable fats, that would be helpful.

Response 2: We thank the reviewer for asking about clarification related to clear up the ambiguity of blended and average oils meaning. Actually, in Moroccan gastronomy, we often use mixed vegetable oils for cooking. We have specified the mixture most used in Moroccan cooking in lines 230 and 231.

Considering the reviewer’s comment, and since vegetable fats means margarines, we left only the term “margarines” in the manuscript. Please see line 236.

Point 3: Lines 262-263: Not all readers are familiar with these abbreviated region names. Please spell out or add more common names.

Response 3: Thank you very much for your valuable suggestion. We have added the meaning of the abbreviations. Please see lines 273-276.

Point 4: Lines 309-321: Another potential source of nutrients is supplements. Please comment on the prevalence of supplement use in this region and on the decision to not collect this data.

Response 4: Thank you very much for this constructive comment about supplements. In actual fact, while the use of supplements is common in the most of developed countries; it is very rare in the case of Morocco. The validated FFQ used in the present study takes into account food supplements (vitamins, selenium, and others), and this type of contribution is very negligible in Moroccan food habits. In this study, for example, no case reported using dietary supplements including PUFA intake. In fact, given the wealth of our country in oily fish namely Sardines which is an important source of PUFA, our population does not suffer from a dietary deficiency linked to this nutrient. We have accordingly integrated this information in the result part. Please see line 240.

Point 5: Lines 326-329: FFQ was administered in 2009 to 2017 (lines 109-110). Using updated version of 2020 food composition table (lines 326-329) may not reflect what study participants have actually consumed due to changes in food supply. Please justify the use of 2020 in your study. Please see reference.

Response 5: Thank you so much for raising this concern. In fact, the previous and first Moroccan food composition table (FCT) entitled “Food composition Table for use in Morocco” was published in 1977, by the Ministry of Agriculture of Morocco (El Khayate, 1984) and revised in 1984 by El Khayate R (El Khayate, 1984). Please see the reference below:

Khalis M, Garcia-Larsen V, Charaka H, Sidi Deoula MM, El Kinany K, Benslimane A, et al. Update of the Moroccan food composition tables: Towards a more reliable tool for nutrition research. Journal of Food Composition and Analysis. 1 avr 2020;87:103397.

https://www.sciencedirect.com/science/article/abs/pii/S0889157519301280

This old version of FCT does not take into account dietary changes during the last two decades in Morocco, and does not include several foods which were introduced in the updated version, and which represent an addition of 79 % of foods in the FCT. We acknowledge that changes in food supply may occur during time. However, we estimate that the closed FCT to what study participants have actually consumed is the updated version. To avoid any ambiguity, this justification was added in the manuscript. Please see lines from 350-354.

Round 2

Reviewer 2 Report

Thank you for answering my comment.

However, I feel that the reply to my previous comment 3 is not enough.

How did you decide on the PUFA cutoff value of 16.87g/day in this answer?

And the authors show in Table 3 that the difference in energy intake between the Kras + group and the Kras- group is significantly large. I agree with the author's reply, "The high energy intake is mainly due to carbohydrates." And although I could understand the author's reply, "carbohydrates do not constitute a risk factor for CRC or mutations either," I completely disagree. This is because high carbohydrate intake probably also affects blood glucose levels and can cause systemic metabolic fluctuations such as insulin resistance and the generation of reactive oxygen species. I don't think it's fair to focus solely on differences in PUFA intake in this study.

Author Response

Response to Academic Editor Notes

Point 1: Line 77: "legume" for "legumes"
Line 95: "which can induces oxidative". Lose the "s" in "induces"
Line 304: Lose "the" in "...development of the colorectal cancer nor..."

Response 1: We highly appreciate the editor’s helpful comment aiming to improve the quality of manuscript. The typos were corrected as suggested by the editor.

Response to Reviewer 2 Comments

Point 1: How did you decide on the PUFA cutoff value of 16.87g/day in this answer?

Response 1: We thank the reviewer for asking about clarification related to The PUFA cutoff value of 16.87g/day. In fact, this cutoff value represents the median intake.

Point 2: The authors show in Table 3 that the difference in energy intake between the Kras + group and the Kras- group is significantly large. I agree with the author's reply, "The high energy intake is mainly due to carbohydrates." And although I could understand the author's reply, "carbohydrates do not constitute a risk factor for CRC or mutations either," I completely disagree. This is because high carbohydrate intake probably also affects blood glucose levels and can cause systemic metabolic fluctuations such as insulin resistance and the generation of reactive oxygen species.

Response 2: We would like to thank the reviewer for raising this important point which still sparks a hate debate about the association between carbohydrate intake and colorectal cancer. We fully agree with the reviewer’s remark that higher dietary carbohydrate intake results in hyperglycaemia and hyperinsulinaemia; which may further induce the carcinogenesis of colorectal cancer. However, the association between dietary carbohydrate intake and colorectal cancer risk remains controversial. According to the WCRF colorectal cancer report which was revised in 2018, carbohydrates were not identified as a risk factor for CRC (Please see reference number 1). In addition, in the Multiethnic Cohort Study conducted to determine the risk of colorectal cancer associated with glycemic load, carbohydrate, and sucrose, the data from 2379 incident cases of colorectal adenocarcinoma revealed that glycemic load and carbohydrate intake appear to protect against colorectal cancer in women (Please see reference number 2).

Reference number 1: World Cancer Research Fund Network. Diet, nutrition, physical activity and colorectal cancer. Revised 2018.

Reference number 2: Nancy C Howarth , Suzanne P Murphy, Lynne R Wilkens, Brian E Henderson, Laurence N Kolonel. The association of glycemic load and carbohydrate intake with colorectal cancer risk in the Multiethnic Cohort Study. Am J Clin Nutr. 2008 Oct;88(4):1074-82.

https://pubmed.ncbi.nlm.nih.gov/18842796/

Following the reviewer’s suggestion, some clarifications have been added to the revised manuscript. Please see lines 301-303.

Point 3: I don't think it's fair to focus solely on differences in PUFA intake in this study.

Response 3: Following the reviewer's remark, a comment was added to the limitations of the study. Kindly see lines 339-342.

We would like to thank the referee again for taking the time to review our manuscript.
We look forward to hearing from you in due time regarding our submission and to respond to any further questions and comments you may have.

Thank you for your consideration.

On behalf of all of the co-authors,

Achraf EL ASRI
